# A Randomized, Double-Blind, Placebo-Controlled Parallel Study on the Efficacy and Safety of *Centella asiatica* L. Extract for Reducing Alanine Transaminase (ALT) Level in Subjects with Elevated ALT

Yong Joon Jeong [1], Hyelin Jeon [1] and Se Chan Kang [2],*

1 Research Institute, GENENCELL Co., Ltd., Yongin 16950, Korea; jeyoon@genencell.co.kr (Y.J.J.); jeonhl0219@genencell.co.kr (H.J.)
2 Department of Oriental Medicine Biotechnology, College of Life Sciences, Kyung Hee University, Yongin 17104, Korea
* Correspondence: sckang@khu.ac.kr; Tel.: +82-31-201-5637

**Featured Application: This study is a human application test of *Centella asiatica* L. extract (CA-HE50) that can be applied as a raw material for healthy functional food.**

**Abstract:** The liver is an important organ that detoxifies various metabolites, synthesizes proteins, and produces biochemicals necessary for life. There are many medications on the market to treat liver diseases, but these can be a strain on the liver due to the need for a detoxification process in the organ. Herbal medicines are replacing synthetic drugs. In the present study, we aimed to evaluate the efficacy and safety of *Centella asiatica* L. extract for reducing alanine transaminase (ALT) levels using a randomized, double-blind, placebo-controlled parallel study. Investigators performed a clinical trial in which an herbal treatment was administered every morning for 12 weeks to 80 patients in two groups. The study protocol number was SYN/RM/CA-008. The results demonstrated improved ALT levels with a positive change in the investigational product (IP) group (−19.9) compared to the placebo group (1.8) ($p < 0.0001$). In addition, IP treatment was safe and non-toxic. The current data indicate that CA-HE50 exhibits clinically significant changes for all hepatoprotective efficacy parameters, suggesting potential for development and applicability as a hepatoprotective substance.

**Keywords:** *Centella asiatica* L.; hepatoprotection; alanine aminotransaminase; human application trial; functional food

## 1. Introduction

The liver is one of the essential organs in the body that regulates various processes. Among them, the functions of metabolism, secretion, storage, and detoxification of endogenous and exogenous substances are essential [1]. Due to its vital function, the liver is important, and disease of the organ constitutes a major threat to public health. Liver disease or damage caused by viral infections, inflammation, toxic substances, and genetics can lead to liver failure [2]. Liver failure refers to a condition in which the liver is unable to perform its normal synthetic and metabolic functions. Types of liver failure include acute and chronic failure [3]. Acute liver failure can be associated with rapidly progressing multi-organ failure and fatal complications. Due to its widespread effects, liver failure can induce systemic inflammatory responses [4], high energy expenditure, and catabolism [5]. In the liver itself, failure can induce loss of metabolic function and decreased synthetic capacity, leading to coagulopathy. For this reason, it is important to prevent and manage liver disease. Unfortunately, there are no effective drugs to stimulate liver function, completely protect organs, or aid in liver-cell regeneration without side effects [6]. Therefore, it

is necessary to find a more effective and non-toxic alternative for preventing and protecting against liver disease.

To solve this issue, research on natural products is being conducted. *Centella asiatica* L., also called gotu kola or pennywort, is a clonal perennial herbaceous creeper belonging to the family *Apiaceae* (*Umbelliferae*) [7]. *Centella asiatica* L. is an important medicinal herb used in the East and gaining popularity in the West [8]. In addition, it has various traditional, medical, and therapeutic values. *Centella asiatica* L. is an ethnomedicinally important plant that is used globally for the treatment of jaundice, hepatitis, syphilis, measles, smallpox, asthma, urethritis, renal stones, rheumatism, varicose veins, neuralgia, anorexia, and skin diseases due to its analgesic, antipyretic, and anti-inflammatory properties [9]. *Centella asiatica* L. has been reported to have anti-inflammatory [10], antioxidant [11–13], wound-healing [14], and memory-enhancing properties [15,16]. In addition, *Centella asiatica* L. has been reported to have hepatoprotective effects, and the mechanism is to influence the increase of antioxidant enzymes and the decrease of inflammatory mediators [17–19]. It was confirmed that CA-HE50, an extract of *Centella asiatica* L., independently developed and standardized by our research team, has antioxidant, anti-inflammatory, and liver-protection effects both in vitro and in vivo. These effects were exerted by asiaticoside, a triterpenoid present in CA-HE50 [20]. Unfortunately, human application trials to evaluate the liver-health efficacy of *Centella asiatica* L. have not been conducted. Therefore, based on the results of a previous study [20], we attempted to evaluate the efficacy and safety of CA-HE50 in reducing ALT levels in humans. Based on our results, we confirm the possibility of developing CA-HE50 as a raw material for pharmaceutical treatments and as a functional food.

## 2. Materials and Methods

### 2.1. Research Ethics and Regulatory Approval

Documents related the human application trial, including the research protocol of this study, all revisions, and explanation documents for the test subjects were reviewed and approved by an independent ethics committee. This study was conducted in accordance with the Declaration of Helsinki (52nd World Medical Association General Assembly, Edinburgh, Scotland, October 2000), and this human application trial was conducted in accordance with The International Conference on Harmonization (ICH) guidelines on Good Clinical Practice (GCP), E6 (R2), and ICH-GCP E6 (R2) guidelines. It was carried out in compliance with the requirements of local regulatory agencies, and approval from the Clinical Trial Registry India (CTRI) was obtained before the trial. The study was conducted with approval of the Ethics Committee of Rajalakshmi Hospital (Principal investigator, Dr. Giriraj, Bengaluru, Karnataka, India) on 12 August 2019 and Vagus Super Speciality Hospital (Principal investigator, Dr. Prakash, Bengaluru, Karnataka, India) on 13 August 2019 (IRB # SYN/RM/CA-008). All subjects provided informed consent prior to inclusion in the study.

### 2.2. Patient Information and Consent

All patients provided written informed consent to participate in the study prior to being screened. The patient information sheet detailed the procedures involved in the study (aims, methodology, potential risks, and anticipated benefits), and the investigator explained them to each patient. The patient signed the consent form to indicate that the information had been explained and understood. The patient was allowed time to consider the information presented before signing and dating the informed consent form to indicate that they fully understood the information and willingly participated in the study.

### 2.3. Selection of Patients

This study was conducted to evaluate the efficacy and safety of CA-HE50 compared to placebo for improving alanine aminotransferase (ALT) levels in subjects with high ALT levels. The study proceeded with the schedule and measurement items shown in Table 1.

**Table 1.** Study schedule.

| Parameter | | Visit 1 | Visit 2 | Visit 3 | Visit 4 | Safety F/U |
|---|---|---|---|---|---|---|
| | | −2 Weeks Day−14~−1 | 0 Weeks Day 0 ± 5 | 6 Weeks Day 42 ± 5 | 12 Weeks Day 84 ± 5 | 12~14 Weeks (If Applicable) |
| Subjects consent | | ● | | | | |
| Demographic survey | | ● | | | | |
| Physical examination | | ● | | | | |
| Vital signs | | ● | ● | ● | ● | |
| Medical history and drug investigation | | ● | | | | |
| Combination therapy investigation | | ● | ● | ● | ● | |
| Pregnancy test (if applicable) | | ● | ● | ● | ● | |
| Conformity assessment | | ● | | | | |
| Randomization | | | ● | | | |
| Effectiveness evaluation | ALT | | ● | | ● | |
| | AST | | ● | | ● | |
| | GGT | | ● | | ● | |
| | Triglyceride | | ● | | ● | |
| | Total cholesterol | | ● | | ● | |
| | LDL-cholesterol | | ● | | ● | |
| | HDL-cholesterol | | ● | | ● | |
| Safety evaluation | Vital signs | | ● | | ● | |
| | Blood biochemical test | | ● | | ● | |
| | Hematological test | | ● | | ● | |
| | Urinalysis | | ● | | ● | |
| Food prescription for human application test | | | ● | ● | | |
| Collection of returned product | | | | ● | ● | |
| Investigation of adverse events | | | | ● | ● | ● |

### 2.3.1. Inclusion Criteria

A total of 80 subjects was selected to participate in this study. Inclusion criteria were:

◆ adult men and women aged 20 to less than 70 years
◆ ALT level 45 ≤ or < 135 U/L
◆ an aspartate aminotransferase (AST) level of 25 U/L or higher in the previous two weeks
◆ no serious liver-related disease with mild liver dysfunction
◆ those who could follow the dietary guidelines of the trial plan during the participation period
◆ those who could comply with the dates of the visits during the participation period
◆ persons who agreed to and signed written informed consent.

### 2.3.2. Exclusion Criteria

Exclusion criteria were as follows:

◆ persons who experienced hypersensitivity or allergic reactions to the test/control foods or ingredients
◆ people with liver-related disease (hepatitis, cirrhosis, alcoholic liver disease, drug-induced liver disease)
◆ people whose levels of ALT/AST/gamma-glutamyl transferase (GGT) exceeded the normal range by three times
◆ people with chronic alcoholic disease—30 g or more of alcohol consumed per day (120 g/week)

◆ people with systemic diseases (hypercholesterolemia, kidney disease, diabetes, etc.)
◆ people with a cancer-related treatment history/hepatobiliary disease or jaundice history within the previous five years
◆ pregnant and lactating women
◆ people with alcohol or drug abuse experience
◆ people who consumed prescription drugs related to weight or appetite within the previous four weeks or who continuously consumed drugs that might affect liver function, such as herbal medicines and supplements
◆ persons with a history of clinically significant cardiovascular disease within the previous six months
◆ people who had been involved in other clinical trials within the previous three months
◆ people who were judged by the researcher to be inappropriate to participate in this test (the researcher excluded subjects if it was determined that it would be detrimental to their safety or welfare).

### 2.4. Experimental Conditions

2.4.1. Preparation of the Investigational Product

*Centella asiatica* L. was purchased from a plantation in Hapcheon (Gyeongsangnam-do, Korea) in August 2017. The specimen was identified by Professor Kang from Kyung Hee University (Yongin, Gyeonggi-do, Korea), and a voucher specimen (JBR536) was deposited in the Laboratory of Natural Medicine Resources at the BioMedical Research Institute, Kyung Hee University. The preparation method of *Centella asiatica* L. extract was the same as described by Park et al. [21]. *Centella asiatica* L. was dried while avoiding direct sunlight, pulverized, and extracted with 50% ethanol for 8 h at 80 °C. The extract was concentrated to 20–25 Brix at reduced pressure and 65 °C in a rotary evaporator, spray-dried to obtain powder, and stored at −20 °C until use. *Centella asiatica* L. 50% ethanol extract was named CA-HE50. The content of asiaticoside, an active component of CA-HE50, was analyzed through high-performance liquid chromatography [21]. The investigational-product (IP) tablet and placebo tablet were produced by COSMAXBIO (Seongnam, Gyeonggi-do, Korea), a good manufacturing product (GMP) factory in Korea, and the compositions are shown in Table 2.

**Table 2.** Ingredients of the IP and placebo.

| Investigation Product | | Placebo | |
|---|---|---|---|
| Ingredients | Mg Per Unit (Mg) | Ingredients | Mg Per Unit (Mg) |
| *Centella asiatica* L. extract (CA-HE50) | 300.0006 | Maltodextrin | 270.0000 |
| Microcrystalline cellulose | 321.6609 | Microcrystalline cellulose | 450.0000 |
| Milk sugar, lactose | 225.0000 | Milk sugar, lactose | 126.6615 |
| Silica dioxide | 13.5000 | Silica dioxide | 13.5000 |
| Magnesium stearate | 9.0000 | Magnesium stearate | 9.0000 |
| Hydroxypropylmethyl cellulose (HPMC) | 21.6000 | Hydroxypropylmethyl cellulose (HPMC) | 21.6000 |
| Glycerin Esters of Fatty Acids | 2.1600 | Glycerin Esters of Fatty Acids | 2.1600 |
| Titanium dioxide | 4.6800 | Titanium dioxide | 4.6800 |
| Gardenia yellow | 1.7500 | Gardenia yellow | 1.7550 |
| Gardenia blue | 0.6435 | Gardenia blue | 0.6435 |
| **Total** | 900.0000 | **Total** | 900.0000 |

2.4.2. Experimental Procedure

A randomized, double-blind, placebo-controlled, parallel-design, 12-week intake study was conducted to evaluate the efficacy and safety of the test substance CA-HE50 for improving ALT levels in subjects. A total of 80 subjects was divided into two groups of 40 subjects each. Group A took one tablet of the investigational product (IP), while Group B took one tablet of the placebo every morning for 12 weeks (84 days). Simple randomization

was performed. The principal investigator (PI) followed the randomizations to maintain blinding of all the people involved in the research, and the blinding code was verified according to the protocol. The grouping and disposition of the subjects are shown in Figure 1.

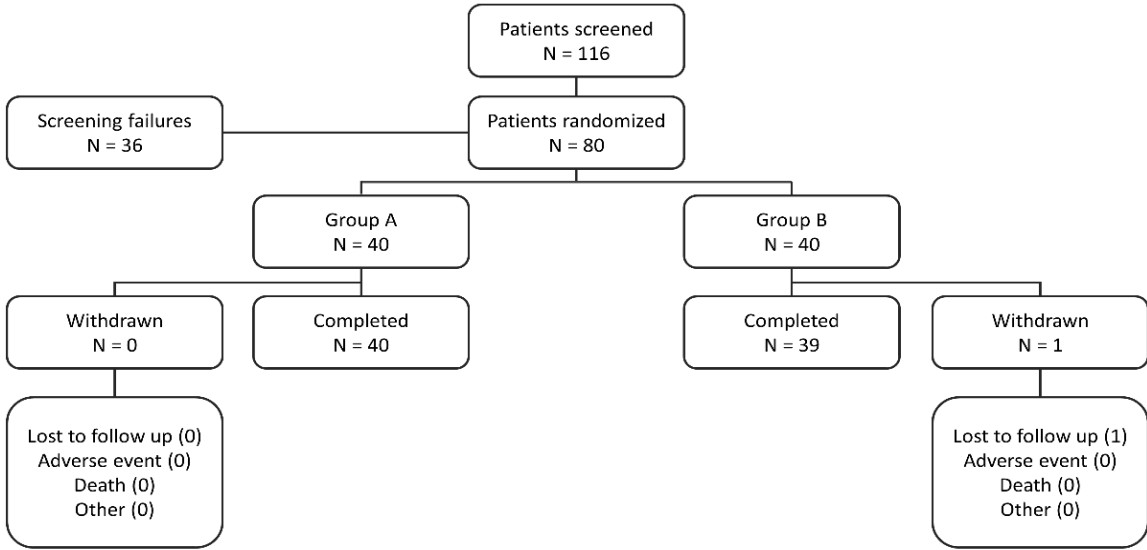

**Figure 1.** Flow diagram describing the participants of the study.

This human application trial was planned to comprise a total of 4 visits, including screening tests and final visits, and additional visits were performed at the request of the study subjects or when the researcher deemed it necessary for reasons such as follow-up on adverse reactions. Subjects were informed by the researcher that they had the right to withdraw their participation in the study at any time without providing a reason. In addition, the investigator could withdraw subjects from the trial if they deemed it appropriate for safety or ethical reasons or if it was considered to be detrimental to the wellbeing of the subjects. Researchers trained subjects to adhere to dietary guidelines that did not affect their liver function during the trial (such as not taking medications that can affect liver function, such as herbal medicines, supplements, or appetite supplements, and not abusing alcohol or drugs). Meanwhile, the researchers ensured hospital visits by calling several days before the scheduled visit date to prevent loss of the subjects.

### 2.5. Validity-Evaluation Parameters

Alanine aminotransferase (ALT), aspartate transaminase (AST), gamma-glutamyl transferase (GGT), triglyceride (TG), total cholesterol (TC), low-density lipoprotein cholesterol (LDL), and high-density lipoprotein cholesterol (HDL) were measured. The change between baseline (visit 2) and end of treatment (EOT, visit 4) was evaluated.

### 2.6. Safety-Evaluation Parameters

◆ Vital signs: body temperature, systolic blood pressure, diastolic blood pressure, and pulse rate
◆ Blood chemistry tests of total bilirubin (T-BIL), alkaline phosphatase (ALP), total protein, albumin (ALB), blood urea nitrogen (BUN), creatinine (CRE), glucose (GLU), and uric acid, as well as hematological tests of hemoglobin (Hb), hematocrit (HCT), red blood cells (RBC), and platelets
◆ Urine test of color, appearance, sugar, protein, pH, specific gravity, ketone bodies, urobilinogen, bile salts, bile pigments, and microscopic examination (pus cells, epithelial cells, RBCs, crystals, casts, and bacteria)
◆ Adverse events (AE) were measured.

The amount of change between baseline (visit 2) and EOT (visit 4) of the above parameters was evaluated.

### 2.7. Data Quality Assurance

The study was conducted based on currently approved protocols and relevant standard operating procedures (SOP). Regular monitoring was performed as per the monitoring plan. All the data recorded in the case report form (CRF) were evaluated for accuracy, credibility, consistency, and quality with source data. All data were captured based on ICH-GCP guidelines and applicable regulations.

### 2.8. Statistical and Analytical Plans

Continuous evaluation variables are represented by arithmetic mean (standard deviation)/geometric mean (coefficient of variation), while categorical variables are represented by numbers (n) and percentages (%). Demographic variables and baseline values were calculated in the full analysis set (FAS) and safety set. In addition, the efficacy variable was analyzed using the FAS, and the safety variable was calculated in the safety set.

Effectiveness variables were expressed as the value of visit 2, the value of visit 4, and the amount of change between visit 2 and visit 4. The comparison between groups was performed using independent *t*-tests, and the comparison between visit 2 and visit 4 within each group was performed using the corresponding *t*-test. When the *p*-value of the *t*-test was significant, a normality test was performed; when normality was satisfied in each group, the *t*-test result was considered significant ($p < 0.05$). When normality was not satisfied in at least one group, the Mann-Whitney U test was performed for the corresponding *t*-test, and a Wilcoxon's rank-sum test was performed for the independent *t*-test. When there was a significant difference between groups at baseline, analysis of covariance (ANCOVA) was performed using the baseline value as a covariate. The safety set was used in the safety variable analysis, and missing data were not replaced.

## 3. Results

The subject demographics at baseline are shown in Table 3. There were no significant differences in age, height, weight, body mass index (BMI), temperature, systolic blood pressure (SBP)/diastolic blood pressure (DBP), and pulse mean values between the two groups.

**Table 3.** Summary of the demographics and continuous parameters in all enrolled subjects.

| Statistics | CA-HE50 | Placebo | *p*-Value |
|---|---|---|---|
| **Age (Years)** | | | |
| N | 40 | 40 | |
| Mean | 37.7 | 36.2 | >0.05 |
| SD | 12.2 | 10.7 | |
| Median | 37.0 | 34.0 | |
| Range (Min, Max) | (20.0, 70.0) | (20.0, 59.0) | |
| **Height (cm)** | | | |
| N | 40 | 40 | |
| Mean | 166.9 | 164.3 | >0.05 |
| SD | 11.1 | 11.8 | |
| Median | 165.0 | 165.0 | |
| Range (Min, Max) | (145.0, 185.0) | (134.0, 185.0) | |
| **Weight (kg)** | | | |
| N | 40 | 40 | |
| Mean | 66.4 | 63.8 | >0.05 |
| SD | 10.8 | 12.3 | |
| Median | 65.0 | 62.5 | |
| Range (Min, Max) | (48.0, 96.0) | (40.0, 94.0) | |

**Table 3.** *Cont.*

| Statistics | CA-HE50 | Placebo | *p*-Value |
|---|---|---|---|
| **BMI (kg/m²)** | | | |
| N | 40 | 40 | |
| Mean | 23.8 | 23.6 | >0.05 |
| SD | 2.8 | 3.8 | |
| Median | 23.5 | 23.7 | |
| Range (Min, Max) | (19.5, 32.1) | (15.0, 36.7) | |
| **Temperature (Celsius)** | | | |
| N | 40 | 40 | |
| Mean | 36.9 | 36.9 | >0.05 |
| SD | 0.1 | 0.1 | |
| Median | 37.0 | 37.0 | |
| Range (Min, Max) | (36.5, 37.0) | (36.5, 37.0) | |
| **SBP (mmHg)** | | | |
| N | 40 | 40 | |
| Mean | 116.2 | 117.3 | >0.05 |
| SD | 5.8 | 5.5 | |
| Median | 118.0 | 120.0 | |
| Range (Min, Max) | (110.0, 130.0) | (110.0, 130.0) | |
| **DBP (mmHg)** | | | |
| N | 40 | 40 | |
| Mean | 76.4 | 77.8 | >0.05 |
| SD | 6.4 | 5.4 | |
| Median | 78.0 | 80.0 | |
| Range (Min, Max) | (60.0, 90.0) | (70.0, 90.0) | |
| **Pulse (BPM)** | | | |
| N | 40 | 40 | |
| Mean | 70.2 | 71.2 | >0.05 |
| SD | 9.1 | 8.9 | |
| Median | 70.0 | 72.0 | |
| Range (Min, Max) | (56.0, 88.0) | (58.0, 100.0) | |

*3.1. Liver-Protection Effectiveness Evaluation Parameter Result*

We selected ALT, AST, GGT, TG, TC, LDL, and HDL as the efficacy-evaluation variables measured at visit 2 and visit 4. In each group, the difference between the intake endpoint and baseline was analyzed, and the two groups were compared.

Group A subjects exhibited a clinically significant change at EOT compared to baseline (Table 4). For Group A, the mean change of ALT between visit 2 and visit 4 for 12 weeks was $-19.9$ ($p < 0.0001$). For Group B, mean change between visit 2 and visit 4 was $-1.8$, which was not statistically significant. The mean change in Group A for AST was $-11.4$, which was statistically significant ($p < 0.0001$), but the mean change in Group B was +0.4, exhibiting little change. The mean change in GGT was $-6.8$ ($p < 0.0001$) within Group A and $-1.5$ ($p < 0.05$) within Group B.

Furthermore, as shown in Table 5, the mean change in TG was $-18.3$ ($p < 0.001$) in Group A and +5.7 (no significant difference) in Group B. The TC measurement also exhibited clinically meaningful results. The mean change in Group A was $-8.9$ ($p < 0.01$), while that of Group B was +16.3 ($p < 0.0005$), revealing opposite results for the two groups. The average change in LDL cholesterol (when comparing visit 2 to visit 4) was +3.6 (no significant difference) in Group A and +19.7 ($p = 0.0001$) in Group B, but the mean change in Group A HDL cholesterol was $-1.7$ ($p < 0.01$), while that of Group B was +3.0 ($p < 0.005$). A summary of these results is shown in Table 6.

**Table 4.** Summary of change from baseline to determine the liver-protection effectiveness evaluation parameter of liver enzymes.

| ALT | | Baseline (Visit 2) | | | | Visit 4 | | | | Change from Baseline | | | | Within Group † | Within Group ‡ | Between Group * |
|---|---|---|---|---|---|---|---|---|---|---|---|---|---|---|---|---|
| Group | N * | Mean | SD | Median | (Min, Max) | Mean | SD | Median | (Min, Max) | Mean | SD | Median | (Min, Max) | | | |
| CA-HE50 | 40 | 54.3 | 9.5 | 50 | (45, 91) | 34.4 | 13.2 | 35 | (12, 69) | −19.9 | 9.9 | −21 | (−42, 1) | <0.0001 | <0.0001 | <0.0001 |
| Placebo | 39 | 55.9 | 9.9 | 53 | (45, 85) | 54.2 | 14.7 | 55 | (17, 90) | −1.8 | 11.3 | 1 | (−39, 8) | 0.3323 | 0.2891 | |

| AST | | Baseline (Visit 2) | | | | Visit 4 | | | | Change from Baseline | | | | Within group † | Within group ‡ | Between group * |
|---|---|---|---|---|---|---|---|---|---|---|---|---|---|---|---|---|
| Group | N * | Mean | SD | Median | (Min, Max) | Mean | SD | Median | (Min, Max) | Mean | SD | Median | (Min, Max) | | | |
| CA-HE50 | 40 | 42.3 | 17.9 | 35 | (26, 91) | 30.9 | 13.8 | 27 | (14, 77) | −11.4 | 9.8 | −11 | (−32, 9) | <0.0001 | <0.0001 | <0.0001 |
| Placebo | 39 | 44.2 | 19.4 | 36 | (25, 99) | 44.6 | 18.4 | 39 | (14, 95) | 0.5 | 5.1 | 0 | (−17, 12) | 0.6388 | 0.3940 | |

| GGT | | Baseline (Visit 2) | | | | Visit 4 | | | | Change from Baseline | | | | Within group † | Within group ‡ | Between group * |
|---|---|---|---|---|---|---|---|---|---|---|---|---|---|---|---|---|
| Group | N * | Mean | SD | Median | (Min, Max) | Mean | SD | Median | (Min, Max) | Mean | SD | Median | (Min, Max) | | | |
| CA-HE50 | 40 | 36.2 | 14.4 | 34 | (15, 65) | 29.4 | 12.3 | 27 | (14, 62) | −6.8 | 6.3 | −5 | (−25, 1) | <0.0001 | <0.0001 | <0.0001 |
| Placebo | 39 | 38.2 | 17.5 | 33 | (11, 72) | 36.7 | 16.3 | 30 | (14, 70) | −1.5 | 4.2 | −2 | (−12, 9) | 0.0271 | 0.0202 | |

Only subjects with a non-missing value for ALT, AST, and GGT at baseline at visit 4 were considered. † Within group comparison was performed using paired *t*-test. ‡ Within group comparison was performed using a non-parametric method (Wilcoxon signed-rank test). * Between group comparison was performed using two-sample *t*-test that compared IP and placebo. Alanine aminotransferase, ALT; aspartate aminotransferase, AST; gamma-glutamyl transferase, GGT.

**Table 5.** Summary of change from baseline to determine the– liver protection effectiveness evaluation parameter of lipid profile.

| TG | | Baseline (Visit 2) | | | | Visit 4 | | | | Change from Baseline | | | | Within Group † | Within Group ‡ | Between Group * |
|---|---|---|---|---|---|---|---|---|---|---|---|---|---|---|---|---|
| Group | N * | Mean | SD | Median | (Min, Max) | Mean | SD | Median | (Min, Max) | Mean | SD | Median | (Min, Max) | | | |
| CA-HE50 | 40 | 173.1 | 82.1 | 153 | (70, 496) | 154.9 | 72.6 | 136 | (60, 399) | −18.3 | 31.5 | −6 | (−108, 52) | 0.0007 | 0.0004 | 0.0008 |
| Placebo | 39 | 169.2 | 78.0 | 146 | (86, 452) | 174.9 | 84.7 | 150 | (93, 503) | 5.7 | 29.7 | 2 | (−56, 90) | 0.2362 | 0.3221 | |

| TC | | Baseline (Visit 2) | | | | Visit 4 | | | | Change from Baseline | | | | Within group † | Within group ‡ | Between group * |
|---|---|---|---|---|---|---|---|---|---|---|---|---|---|---|---|---|
| Group | N * | Mean | SD | Median | (Min, Max) | Mean | SD | Median | (Min, Max) | Mean | SD | Median | (Min, Max) | | | |
| CA-HE50 | 40 | 169.4 | 22.7 | 168 | (106, 225) | 160.5 | 29.6 | 164 | (104, 226) | −8.9 | 19.2 | −6 | (−62, 29) | 0.0055 | 0.0028 | <0.0001 |
| Placebo | 39 | 165.7 | 20.9 | 170 | (123, 204) | 182.1 | 29.8 | 177 | (130, 251) | 16.3 | 25.5 | 9 | (−40, 97) | 0.0003 | 0.0001 | |

| LDL | | Baseline (Visit 2) | | | | Visit 4 | | | | Change from Baseline | | | | Within group † | Within group ‡ | Between group * |
|---|---|---|---|---|---|---|---|---|---|---|---|---|---|---|---|---|
| Group | N * | Mean | SD | Median | (Min, Max) | Mean | SD | Median | (Min, Max) | Mean | SD | Median | (Min, Max) | | | |
| CA-HE50 | 40 | 88.6 | 23.9 | 96 | (20, 119) | 92.2 | 26.3 | 96 | (15, 145) | 3.6 | 25.1 | −3 | (−44, 75) | 0.3738 | 0.5692 | 0.0100 |
| Placebo | 39 | 87.1 | 23.7 | 91 | (29, 126) | 106.8 | 25.8 | 105 | (65, 180) | 19.7 | 29.1 | 13 | (−40, 105) | 0.0001 | 0.0001 | |

| HDL | | Baseline (Visit 2) | | | | Visit 4 | | | | Change from Baseline | | | | Within group † | Within group ‡ | Between group * |
|---|---|---|---|---|---|---|---|---|---|---|---|---|---|---|---|---|
| Group | N * | Mean | SD | Median | (Min, Max) | Mean | SD | Median | (Min, Max) | Mean | SD | Median | (Min, Max) | | | |
| CA-HE50 | 40 | 43.8 | 6.4 | 42 | (25, 58) | 42.1 | 5.8 | 42 | (24, 51) | −1.7 | 3.8 | −1 | (−11, 3) | 0.0075 | 0.0290 | <0.0001 |
| Placebo | 39 | 44.9 | 7.5 | 43 | (29, 73) | 47.9 | 8.5 | 48 | (32, 86) | 3.0 | 6.1 | 2 | (−9, 20) | 0.0038 | 0.0032 | |

Only subjects with a non-missing value for the GT, TC, LDL, and HDL at baseline and visit 4 were considered. † Within-group comparison was performed using paired *t*-test. ‡ Within-group comparison was performed using a non-parametric method (Wilcoxon signed-rank test). * Between-group comparison was performed using a two-sample *t*-test that compared IP and placebo. TG, triglyceride; TC, total cholesterol; LDL, low-density lipoprotein cholesterol; HDL, high-density lipoprotein cholesterol.

**Table 6.** Summary of the parameter changes from baseline.

| Parameters | Groups | Change Form Baseline | | | | | | | |
|---|---|---|---|---|---|---|---|---|---|
| | | Mean | SD | Median | Min | Max | Within Group [†] | Within Group [‡] | Between Group * |
| **ALT** | CA-HE50 | −19.9 | 9.9 | −21 | −42 | 1 | <0.0001 | <0.0001 | <0.0001 |
| | Placebo | −1.8 | 11.3 | 1 | −39 | 8 | 0.3323 | 0.2891 | |
| **AST** | CA-HE50 | −11.4 | 9.8 | −11 | −32 | 9 | <0.0001 | <0.0001 | <0.0001 |
| | Placebo | 0.4 | 5.1 | 0 | −17 | 12 | 0.6388 | 0.3940 | |
| **GGT** | CA-HE50 | −6.8 | 6.3 | −5 | −25 | 1 | <0.0001 | <0.0001 | <0.0001 |
| | Placebo | −1.5 | 4.2 | −2 | −12 | 9 | 0.0271 | 0.0202 | |
| **TG** | CA-HE50 | −18.3 | 31.5 | −6 | −108 | 52 | 0.0007 | 0.0004 | 0.0008 |
| | Placebo | 5.7 | 29.7 | 2 | −56 | 90 | 0.2362 | 0.3221 | |
| **TC** | CA-HE50 | −8.9 | 19.2 | −6 | −62 | 29 | 0.0055 | 0.0028 | <0.0001 |
| | Placebo | 16.3 | 25.5 | 9 | −40 | 97 | 0.0003 | 0.0001 | |
| **LDL-C** | CA-HE50 | 3.6 | 25.1 | −3 | −44 | 75 | 0.3738 | 0.5692 | 0.0100 |
| | Placebo | 19.7 | 29.1 | 13 | −40 | 105 | 0.0001 | 0.0001 | |
| **HDL-C** | CA-HE50 | −1.7 | 3.8 | −1 | −11 | 3 | 0.0075 | 0.0290 | <0.0001 |
| | Placebo | 3 | 6.1 | 2 | −9 | 20 | 0.0038 | 0.0032 | |

[†] Within-group comparison was performed using paired *t*-test. [‡] Within-group comparison was performed using a non-parametric method (Wilcoxon signed-rank test). * Between-group comparison was performed using a two-sample *t*-test that compared IP and placebo. ALT, alanine aminotransferase; AST, aspartate aminotransferase; GGT, gamma-glutamyl transferase; TG, triglyceride; TC, total cholesterol; LDL, low-density lipoprotein cholesterol; HDL, high-density lipoprotein cholesterol.

### 3.2. Safety-Evaluation Parameter Results

There were no clinically significant findings in the safety evaluation. During the trial, four subjects reported AEs, of which three were in the IP group and one was in the placebo group. The researcher evaluated AEs that included nausea, tiredness, and fever, and these responses were not associated with the test substances (Table 7). No other serious or significant adverse reactions were reported, and no deaths occurred. At determination of IP ingestion, there were no clinically significant findings in the safety endpoints.

**Table 7.** Summary of adverse events.

| Subject ID | AE No. | Description | Start Date | Ongoing | End Date | Severity | Outcome | Serious | Related to Drug |
|---|---|---|---|---|---|---|---|---|---|
| LFH-VH-002 | 1 | Nausea | 31 August 2019 | N/A | 13 September 2019 | Mild | Recovered with sequelae | No | Unrelated |
| | 2 | Tiredness | 31 August 2019 | N/A | 13 September 2019 | Mild | Recovered with sequelae | No | Unrelated |
| LFH-VH-003 | 1 | Tiredness | 29 August 2019 | N/A | 31 August 2019 | Mild | Recovered with sequelae | No | Unrelated |
| LFH-VH-005 | 1 | Nausea | 1 September 2019 | N/A | 8 September 2019 | Mild | Recovered with sequelae | No | Unrelated |
| LFH-VH-025 | 1 | Nausea | 31 August 2019 | N/A | 3 September 2019 | Mild | Recovered with sequelae | No | Unrelated |
| | 2 | Fever | 31 August 2019 | N/A | 3 September 2019 | Mild | Recovered with sequelae | No | Unrelated |

### 4. Discussion

The liver plays various important roles in the human body, such as metabolism, detoxification, storage, and secretion. Acute or chronic liver failure can occur for various reasons. Among them, acute liver failure occurs suddenly from various causes and can exhibit clinical symptoms of severe liver damage [22]. The main causes of liver damage leading to liver failure show wide geographical variation and depend on liver viral infections and drug use patterns [23,24]. Studies have shown that viral causes dominate in developing countries, including India, with Hepatitis A, B, and E virus infections as common causes [22,25].

In contrast, in many parts of the United States and Western Europe, most cases of liver damage are drug-induced (ex., paracetamol) [26].

We attempted to develop natural-product-based, non-toxic functional raw materials that stimulate liver function, protect organs, or/and foster liver-cell regeneration. Hepatoprotective fruits include grapefruit, blueberries/cranberries, and grapes, while plants include Nopal (Cactus pear) and tuna (Cactus pear fruit), chamomile, silymarin, and blue-green algae spirulina. The foods reduce blood levels of biomarkers that can confirm liver toxicity, such as ALT, AST, and GGT [2]. Silymarin of the milk thistle, well known for its hepatoprotective properties, is non-toxic at maximum oral doses of 2500–5000 mg/kg. However, gastrointestinal disturbances and mild allergic reactions, including urticaria, nausea, headache, arthralgia, itching, and mild laxative symptoms, have been reported [27]. The natural product *Centella asiatica* L. used in this study demonstrated hepatoprotective efficacy in vitro and in vivo [15], and NOAEL 2000 mg/kg was confirmed in a preclinical safety study (not published). In this human application test, CA-HE50 exhibited a clinically significant change at EOT for all effectiveness variables compared to the placebo. In particular, when comparing the CA-HE50 and placebo groups on ALT, AST, and GGT, the CA-HE50 group had significant difference pre- to post-test ($p < 0.0001$). In addition, a positive change with clinical significance was confirmed for fat metabolism ($p < 0.05$). Meanwhile, although the TC and LDL levels of the placebo group at visit 4 increased compared to those at baseline, they were not considered a significant issue because they were not included at dangerous levels. These changes are believed to be due to the individual's lifestyle. During the trial, minor adverse events were identified in 4 patients, but this was unrelated to the IP or placebo. CA-HE50, a food-derived hepatoprotective material, is a safe material with no side effects and has the advantage of exhibiting sufficient liver protection even at a low dose (300 mg/kg/day). It is also expected to be an alternative hepatoprotective product for people suffering from side effects of other natural hepatoprotective products, including milk thistle.

## 5. Conclusions

In conclusion, after ingestion of CA-HE50 300 mg/day for 12 weeks, significant reduction in ALT, AST, and GGT levels was observed ($p < 0.0001$). In addition, there were no clinically serious abnormalities or significant changes in safety evaluation parameters due to CA-HE50 ingestion. These results are expected to be of great help in the subsequent pharmacological use and development of CA-HE50.

**Author Contributions:** Conceptualization, Y.J.J. and S.C.K.; methodology, Y.J.J. and H.J.; software, H.J.; validation, Y.J.J. and H.J.; formal analysis, Y.J.J. and H.J.; investigation, H.J.; resources, Y.J.J. and H.J.; data curation, H.J.; writing—original draft preparation, Y.J.J., and H.J.; writing—review and editing, Y.J.J., H.J. and S.C.K.; visualization, Y.J.J. and H.J.; supervision, Y.J.J. and S.C.K.; project administration, Y.J.J.; funding acquisition, Y.J.J.; Y.J.J. and H.J. should be considered the joint first authors. All authors have read and agreed to the published version of the manuscript.

**Funding:** This study was supported by the Korean Institute of Planning and Evaluation for Technology in Food, Agriculture, Forestry (IPET) through the High Value-Added Food Technology Development Program, funded by the Ministry of Agriculture, Food and Rural Affairs (MAFRA) (117050-3).

**Institutional Review Board Statement:** The study was conducted according to the guidelines of the Declaration of Helsinki and approved by the Institutional Review Board of the Ethics Committee of Rajalakshmi Hospital and Vagus Super Speciality Hospital on 13 August 2019 (IRB# SYN/RM/CA-008).

**Informed Consent Statement:** Informed consent was obtained from all subjects involved in the study.

**Data Availability Statement:** The data presented in this study are available on request from the corresponding author. All data generated or analyzed during this study are included in the manuscript.

**Acknowledgments:** We thank the volunteers for their efforts that made this study possible, in addition, we thank Prakash, Narayanaswamy, and Murthy of Vagus Hospital and Giriraj of Rajalakshmi Hospital, who controlled all aspects of the clinical trials at the local hospitals in India.

**Conflicts of Interest:** The authors declare no conflict of interest.

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
