# Peer review of "A Randomized, Double-Blind, Placebo-Controlled Parallel Study on the Efficacy and Safety of Centella asiatica L. Extract for Reducing Alanine Transaminase (ALT) Level in Subjects with Elevated ALT"

_applsci, doi:10.3390/app112311498_

Round 1
Reviewer 1 Report
Manuscript can be accepted
Reviewer 2 Report
The manuscript entitled ²A randomized, double-blind, placebo-controlled, parallel study that evaluated the efficacy and safety of Centella asiatica extract (CA-HE50) tablets compared to a placebo tablet for improving alanine transaminase (ALT) levels in subjects with elevated² is focused to possibility of developing CA-HE50 as a raw material for pharmaceutical treatments. The authors provide reasonable and meaningful results that provide a clear description of the efficacy and safety of using C. asiatica extract treatment (CA-HE50) in patients with elevated ALT levels.
I recommend the manuscript for publication after considering the following suggestions which their addressing will fit the manuscript for publication.
- Please indicate how your work is different and important versus other competitive materials.
- I recommend that the authors give a more detailed description of Centella asiatica.
Reviewer 3 Report
Dear authors,
in the letter attached, you will find corrections and suggestions for improving (I hope) your work.
The general impression is that you composed this work too rapidly.

Reviewer 4 Report
The article entitled " A randomized, double-blind, placebo-controlled, parallel study that evaluated the efficacy and safety of Centella asiatica extract (CA-HE50) tablets compared to a placebo tablet for improving alanine transaminase (ALT) levels in subjects with elevated ALT " is an interesting study that is focused on the effect of extracts of Centella asiatica on ameliorating an hepatic pathological status. In particular, for the first time, an human application trial on subjects affected by high levels of ALT was conducted in parallel with a double-blind placebo control.
In my opinion the major point is that in Material and methods section should be added a paragraph not only of the composition of the samples of the extracts of Centella asiatica and of the placebo , i.e the ingredients, as the authors report in table 2 but also of the description of their preparation and the analysis performed on the samples before their use in the clinical trial.
Another minor point is a comment on the increase induced by the placebo treatment on TC level and LDL level as reported in Table 5.
For all these reasons the article must to be reconsidered after major revisions

Reviewer 5 Report
Dear Authors
After reading your manuscript, I can say that it is well-prepared and designed. The Title, Abstract and Introduction are good and according to requirements.
I have the following remarks on your manuscript:
- In table 2, number are with high significant figure that can not be real, but only theoretical amounts. Additionally, nothing is said about the tablet formulation procedure.
- In Table 3, Please correct the Range (Min, Max) for both temperature (Celsius) and SBP
- The discussion part is not really a discussion, it is a repetition of what what have been said in the experimental part. So, it should be rewritten.
- The conclusion is too short and does not summarize the work. I can suggest that the last paragraph of the Discussion part should be in the conclusion.
- Although English is good, but I believe that you should go through the manuscript where some mistakes are present. I here show you some of them: Line 95 should be to evaluate rather than to evaluated. Line 101 80 people were selected ... not 80 people was selected....Line 251 should be that stimulates rather than stimulating. Best regards
Round 2
Reviewer 3 Report
Dear authors,
thank you for having taken into account my suggestion and for submitting explanations were required.
My feedback in respect of your revison work is positive.
Kind regards
Reviewer 4 Report
The article entitled " A randomized, double-blind, placebo-controlled, parallel study that evaluated the efficacy and safety of Centella asiatica extract (CA-HE50) tablets compared to a placebo tablet for improving alanine transaminase (ALT) levels in subjects with elevated ALT " is an interesting study that is focused on the effect of extracts of Centella asiatica on ameliorating an hepatic pathological status. In particular, for the first time, an human application trial on subjects affected by high levels of ALT was conducted in parallel with a double-blind placebo control.
The article has been deeply reviewed and the authors have well answered and clarified all the comments that have been presented.
